# Rivers under Ice: Evaluating Simulated Morphodynamics through a Riffle-Pool Sequence

**Karine Smith** [1,2], **Jaclyn M. H. Cockburn** [1,*] and **Paul V. Villard** [2]

1    Department of Geography, Environment & Geomatics, University of Guelph, Guelph, ON N1G 2W1, Canada
2    GEO Morphix Ltd., Cambellville, ON L0P 1B0, Canada; paulv@geomorphix.com
*    Correspondence: jaclyn.cockburn@uoguelph.ca

**Abstract:** Modeling in ice-covered rivers is limited due to added computational complexity, specifically challenges with the collection of field calibration data. Using River2D, a 2-dimensional hydrodynamic modeling software, this study simulates depth-averaged velocity and shear stress distributions under ice cover and in open-water conditions during varying flow conditions in a small, shallow riffle-pool sequence. The results demonstrated differences in velocity distribution throughout the channel and increases in discharge were found to impact the velocity magnitude under ice cover, while the spatial distribution remained consistent. A recirculating eddy found along the pool's left bank was exacerbated under ice cover, with potential implications for silver shiner habitat suitability. Bed shear stress magnitude did not vary significantly between ice and open water, although the spatial distribution differed notably. Model validation demonstrated success in simulating water depth and velocities, and the shear stress was estimated within a reasonable margin. Using hydrodynamic models provides valuable insight into seasonal changes in velocities and shear stress when ice is present.

**Keywords:** hydrodynamics; channel resilience; riffle-pool; River2D; erosion; deposition; velocity distribution; recirculation





## 1. Introduction

Fluvial processes are studied through field research using direct measurements to increase the understanding of channel hydrodynamics under various conditions with different morphological characteristics [1–3]. However, field measurements are time consuming, and it is challenging to cover larger areas in short periods of field time. Ice cover on rivers exacerbates the challenges in field data collection and increases the complexity associated with channel fluvial processes [3–8]. Increasingly, one-, two- and three-dimensional models are used alongside field studies to understand, simulate, and predict the fluvial processes in rivers to support channel design projects, management, and hazard-mitigation efforts [3,6,9–14]. Specifically, modeling is relied on to simulate, predict, and evaluate ice processes that occur on a river. CRISSP2D, RIVICE, DynaRICE, and MESH-RBM are modeling frameworks that simulate the ice formation on large rivers and predict the location of ice jams in an effort to reduce flooding and infrastructure damage [14–17]. Although efficient in simulating ice formation, it is a challenge for models to incorporate hydrodynamics or morphodynamics within a channel, limiting their use in investigating the impacts of ice on fluvial processes (e.g., velocity and shear stress). Additionally, despite the numerous models available for flow modeling under open-channel conditions, there are few models that consider both ice processes and fluvial dynamics, especially models that handle the parameters in small channels (e.g., less than 1 m deep).

Fluvial processes occurring under ice cover require more investigation, especially in the context of flow distribution, thalweg concentration, and shear stress at the bed [2,5,6,10,12,18]. Enhancements in methods to analyze these processes under ice cover

using instrumentation such as acoustic Doppler velocimeters (ADVs) and acoustic Doppler current profilers (ADCPs) provide inputs for flow modeling under ice cover [2,5,6,11]. Numerical models are powerful tools for simulating river processes using computational methods to solve non-linear equations that describe the hydro-morphodynamics of rivers [19]. Numerical modeling incorporates empirical relationships for roughness, velocity, sediment transport capacity, and shear stress to predict flow characteristics under ice cover [20]. Hydrodynamic modeling continues to develop to address the limitations associated with uncertainties related to fluvial dynamics under ice [7]. Lotsari et al. [6] aimed to improve our understanding of the spatial variation of ice-covered mid-winter flow and the erosion and sedimentation capacity of ice-covered flow as compared to open-water using the two-dimensional (2D), hydrodynamic model River2D. River2D has the capacity to simulate direction and magnitude for time- and depth-averaged velocities and to identify recirculating flow structures [21]. The 2D hydrodynamic model produces results that efficiently compare the differences in the near-bed velocities and the spatial distribution of flow under ice-covered conditions and open-channel conditions for meandering rivers [6]. However, changes in the velocity distribution and shear stress under different flow conditions and the influence of a variable bed morphology were not investigated.

The objectives of this study are to simulate ice-covered velocities and shear stress (magnitudes and distributions) under various flow conditions, and to assess the model's performance in a small, ice-covered riffle-pool sequence (Supplementary Materials Figure S1). For this purpose, River2D was used as it includes ice as an upper boundary layer and models depth-averaged velocity and shear velocity distributions [22]. Two scenarios were employed to conduct direct comparisons and investigate the impacts of ice on flow and shear stress: simulation 1 used field data collected during ice cover in lower flow conditions to calibrate an ice-free simulation, and simulation 2 used field data collected during open water in higher flow conditions to calibrate an ice-covered simulation. Hydrodynamic modeling, such as what is presented here, helps users to confidently evaluate processes that are difficult to measure directly (e.g., under ice-covered conditions and dangerous high flow conditions).

## 2. Study Site and Field Measurements

Sixteen Mile Creek in southern Ontario, Canada, has a total watershed area of 372 km$^2$, and it empties into Lake Ontario (Figure 1). It flows southeast through the city of Milton, and experiences impacts from a variety of land-use types including urban, agriculture (cropping and grazing), and forests. The study reach drains an area of 108 km$^2$ and is a 75 m segment, which is ~20 m at its widest (Figure 1b) and ~1 m at its deepest. Based on the Water Survey of Canada gauging station (02HB005, period of record: 1957–present), located approximately 4.5 km upstream of the study reach, the average winter discharge is 1.35 m$^3$/s, the average spring discharge is 2.3 m$^3$/s, and the average summer discharge is 0.8 m$^3$/s [23]. The 2021 average winter discharge was 1.2 m$^3$/s, the average spring discharge was 2.3 m$^3$/s, and the average summer discharge was 0.8 m$^3$/s [23]). Bed roughness was quantified using roughness lengths ($y_0$), which is defined as the height above the bed where the mean velocity is theoretically 0 m/s [24]. The roughness length estimates along seven cross-sections throughout the study site indicated that in the upstream and downstream, most sections of the study reach exhibit bed roughness features that are typical of riffles, while the lower section along cross-sections 4, 5, and 6 exhibit lower levels of roughness, which is typical of pools (Table 1). Wolman pebble counts determined that all cross-sections had a large range in substrate size but were typical for riffles and pools (Table 1). During site visits, sediment transport was limited to suspended sediment transport, and the bedload was negligible [5].

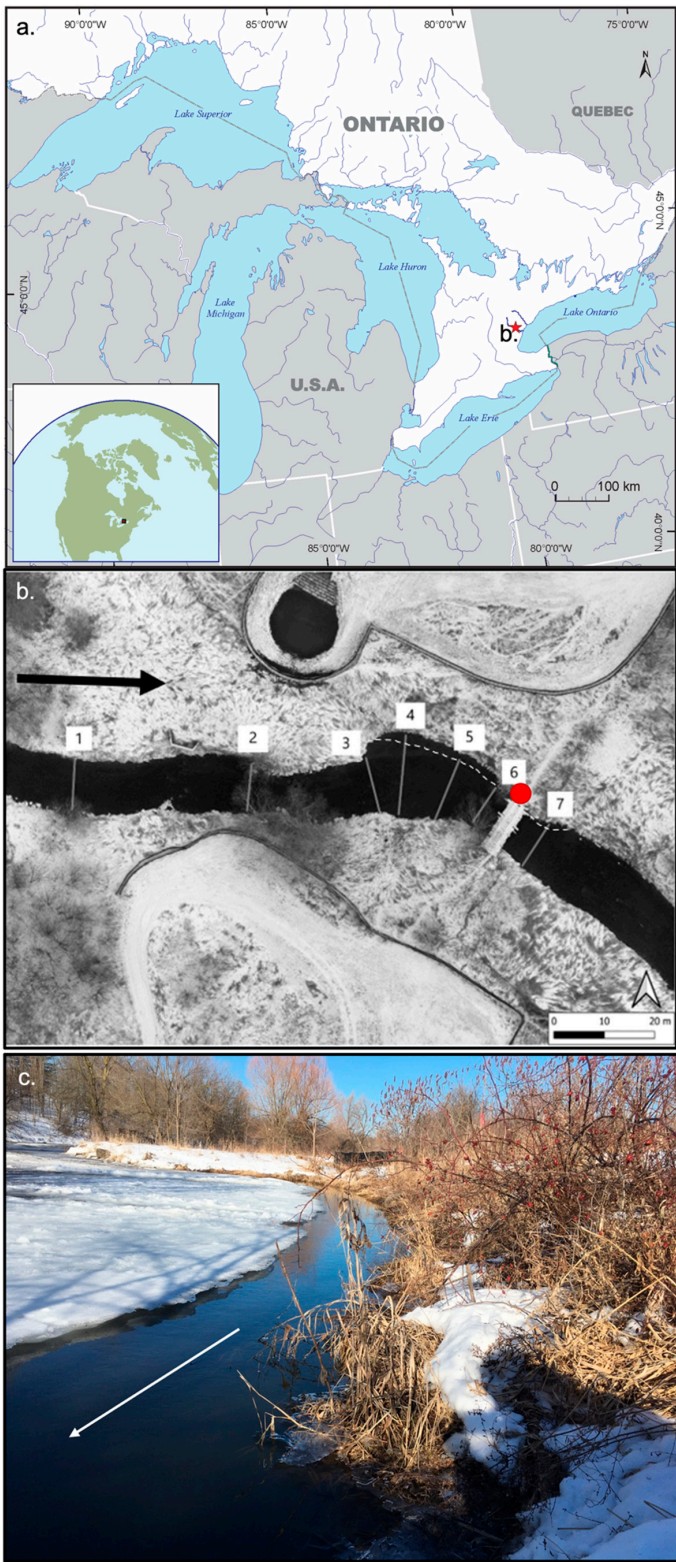

**Figure 1.** Sixteen Mile Creek is a small, low-order stream in southern Ontario, Canada. (**a**) Study reach location relative to the Laurentian Great Lakes and (**b**) aerial photograph showing numbered cross-section locations through the study reach. Flow direction is from left to right, as indicated by the large black arrow along the left bank. The dashed line along the left bank indicates ice extent. (**c**) Photograph looking upstream along the left bank (red dot on (**b**)) showing the thin (~1 m) strip of open water on 25 February 2021 (two days after the major sampling campaign on 23 February 2021). The white arrow in the photograph indicates flow direction.

**Table 1.** Channel geometry and bed substrate characteristics.

| Cross-Section | Width | Average Depth | d/D | Median Grain Size (D50) | Roughness Lengths | Notes |
|---|---|---|---|---|---|---|
| 1—Riffle | 12 m | 0.35 m | 0.85 | 412 mm | 2 mm | Large boulders near left bank |
| 2—Riffle | 10 m | 0.45 m | 3 | 150 mm | 2–2.5 mm | Embedded, poorly sorted fine material found between cobles |
| 3—Riffle | 10 m | 0.28 m | 0.95 | 295 mm | 2.4–2.8 mm | Large, flat boulder 3 m from the right bank, ~25% embedded |
| 4—Pool | 18 m | 0.48 m | 10 | 48 mm | 0.5–0.7 mm | Fine material (sand sized and fine) along the left bank, embedded gravel, cobbles, and boulders toward the right bank |
| 5—Pool | 14 m | 0.52 m | 100 | 5.2 mm | 0.5 mm | Dominated by fine-grained bed material (sand-sized and smaller) |
| 6—Pool | 10 m | 0.6 m | 125 | 4.8 mm | 0.4 mm | Dominated by fine-grained bed material (sand-sized and smaller) |
| 7—Riffle | 9 m | 0.5 m | 1.14 | 439 mm | 2.5–2.7 mm | Cobbles and boulders with embedded pebbles and coarse sand |

Topographic site surveying using a real-time kinematic global position system (RTK GPS) at 3–5 m intervals along the top and bottom of the channel banks, the top of the slope, the edge of the water, and along the channel bed was conducted across the site. Measurements were differentiated based on location using codes: edge of water (EOW), bottom of bank (BOB), top of bank (TOB), channel bed (CB), and central line (CL). Additionally, the ice extent was mapped at 3–5 m intervals throughout the study site on field collection visits [5,25]. Boulders with a b-axis exceeding 0.25 m were surveyed and fed into the elevation model to describe bed heterogeneity throughout [26]. The bed slope was less than 1% through the entire reach (~0.9%), 1.06% through the riffle bed slope, and 1.09% through the pool bed slope. For model calibration and validation purposes, the water levels were continuously monitored from January to June, and the velocity measurements were collected throughout February and March along seven delineated cross-sections (Figure 1) [5,22,25].

The velocity data collected on 23 February 2021 (under ice cover and lower flow conditions) used a Sontek Flowtracker2 Handheld ADV and the Sontek S5 ADCP (where depth was sufficient), and on 2 March 2021 (open water, higher flow conditions), a Sontek S5 ADCP was used. The procedures outlined in Lotsari et al. [7] and Demers et al. [8] were followed on both days. The Flowtracker2 ADV used for this project has an acoustic frequency of 10.0 MHz, a minimum depth requirement of 0.02 m, and depth and velocity resolutions of 0.001 m and 0.0001 m/s [27]. Despite the minimum depth requirement of 0.02 m for the ADV, the measurements could only be completed for depths of 0.05 m or more during this study. The Flowtracker2 ADV also recorded the Signal Noise Ratio (SNR) to aid in detecting measurement errors and/or sensor blockages [27]. The average sampling time was kept relatively short in order to collect data over a wide spatial extent [5]. The S5 ADCP has a minimum depth requirement of 0.2 m and thus is limited in shallow waters. The Sontek S5 has a vertical resolution of ±0.001 m and a velocity resolution of ±0.0001 m/s, and the velocities are averaged between all five transducers [28]. While collecting samples in open water, the ADCP sensor automatically adjusted the velocity cell sizes based on instrument movement speed, water depth, and flow velocity [28]. Sixteen Mile Creek is a shallow channel with a maximum depth of less than a meter, resulting in an average cell size of 0.02 m. The S5 had the blanking distance set at 0.05 m. The average measurement height above the bed was 0.082 m and the average measurement depth below the surface was 0.06 m in open water.

Velocity measurements were collected along all seven cross-sections using a combination of the ADV and ADCP (Figure 1). ADV measurements were collected in 0.05 or 0.10 m-depth increments at 1 m lateral intervals when possible, and at 2 m lateral intervals when time constraints applied. These measurements were taken along each cross-section under both open-water and ice-covered conditions, where holes were drilled through the ice (when present) using augers [7,8]. The ADV was used in combination with the ADCP to collect data in both shallow and deeper locations throughout the channel. Under ice cover,

stationary ADCP measurements were taken through the hole in the ice, from which the flow direction, water depth, and maximum velocity were calculated, and the mean velocity profiles were determined. In ice-free conditions, the ADCP was mounted to a float and manually towed along each cross-section. Where ice was present, the ice thickness and visual descriptions of roughness were also manually measured and recorded.

Since the ADCP records velocities in the East, North, and Up (ENU) directions [29], the E and N vectors were rotated around their axis at an angle corresponding to the streamwise direction to yield u and w velocity components. This was determined using the Velocity Mapping Toolbox (VMT) developed by USGS [29]. The vertical velocity vector was unchanged by the rotation of the E and N velocity rotations. The ADV collects u, v, and w (streamwise, vertical, and lateral) velocities and it did not require additional data processing prior to analysis.

### 3. Pre-Processing, Data Analysis, and Model Validation

The 2-dimensional model River2D was used for hydrodynamic simulations under ice and open water across the study site. River2D requires channel bed and ice surface topography converted into a discrete mesh, roughness estimates, initial flow, and boundary conditions [22]. The boundary conditions were defined as discharge along the inflow boundary, and fixed water surface elevations at the outflow boundary [22]. Depth-averaged velocities were calculated based on the St. Vernant equations: conservation of mass, conservation of x-direction momentum, and conservation of y-direction momentum [22]. A singular velocity value was assigned at each node, and the depth specific velocities were not computed. The model runs on the following three basic assumptions: (1) a hydrostatic pressure distribution across the water column depth, (2) constant distribution of horizontal velocities over depth, and (3) Coriolis and wind forces are negligible [22]. When ice cover was present, River2D used an ice and bed resistance model, where the bed and ice roughness values were combined to create composite roughness used in the hydraulic calculations [22].

The topography data obtained for the ice extent and the high-resolution channel topography survey were converted into Digital Elevation Models (DEMs) using the Triangular Irregular Network (TIN) interpolation tool in River2D_Bed [30]. All topography points were identified as fixed nodes, with elevation and roughness values assigned from the underlying bed topography layer [30]. Hard breaklines were delineated along the deepest location of the channel, as well as the edge of the water and the top of the banks, to prevent 'leakage' from the model [6]. Breaklines ensured the appropriate interpolation of values along linear features in a finalized channel bed elevation TIN (Figure 2) [30]. Throughout winter 2021, full ice cover was present up- and downstream of the pool, with partial cover over the pool, leaving a strip of open water along the left bank (Figures 1c and 2c). The open-water section along the pool is considered within the model by assigning ice thickness values to ice nodes and assigning 0 to no-ice nodes [22]. Breaklines were then established along the ice edge to ensure interpolation along the ice–water boundary. The fixed nodes and breaklines were subsequently smoothed into a mesh element (10 cm resolution) with all the necessary components, allowing the topographical data to be fed into the River2D software [22]. Velocity bins measured using the ACDP under the ice cover ranged from 2 cm–10 cm. The mesh resolution was set to provide sufficient details on velocity and shear stress distribution while still allowing for validation with field data.

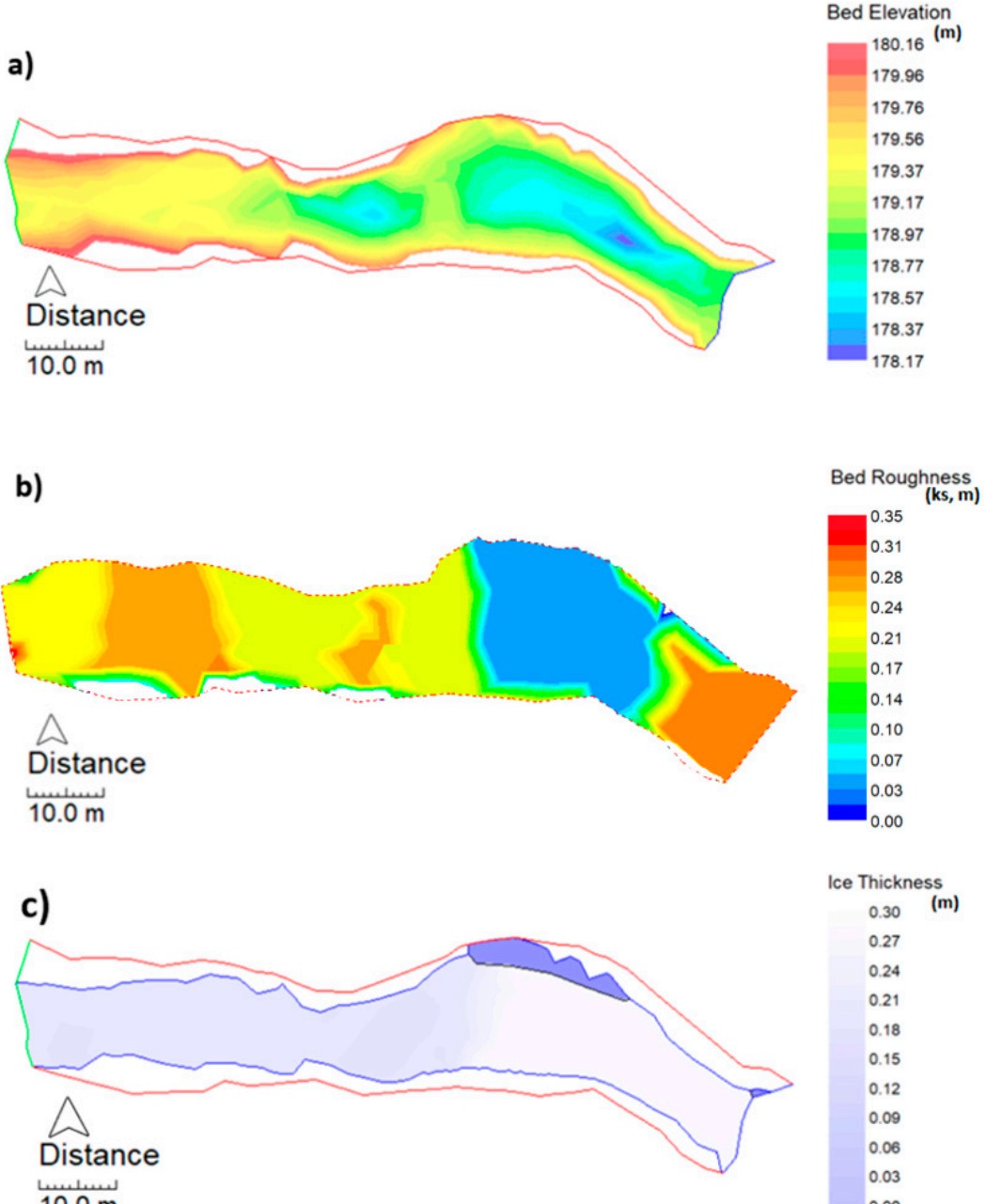

**Figure 2.** Interpolated model surfaces. (**a**) Channel bed elevation, (**b**) channel bed roughness, and (**c**) ice thickness (dark colors represent minimal (or no) ice thickness). Bed roughness was manually prepared in the River2D_bed extension and extends past water edge to ensure roughness values are assigned along the water edge breaklines.

All triangulation points in the mesh element layer require equivalent sand roughness height ($k_s$) estimates, which serve as representations of hydraulic friction across the surface [31]. Roughness heights ($k_s$) were calculated for all measured cross-sections based on estimated roughness lengths from logarithmic velocity profiles seen in Equations (1)–(3):

$$k_s = 30.1 y_0, \tag{1}$$

$$y_0 = e^{-a/b}, \tag{2}$$

$$u_y = a + b \ln y, \tag{3}$$

where $u_y$ is the mean velocity at a given depth, a is the intercept, b is the slope and velocity gradient, and $y_0$ is the calculated roughness length [32]. Individual channel bed nodes throughout the mesh element were assigned roughness height values corresponding to the nearest cross-sections to yield a finalized channel bed roughness TIN. Visual observations and roughness length estimates of the ice surface demonstrated little change in the roughness across the ice surface, allowing for a singular roughness length assigned for the upper boundary ice surface [5,25]. The River2D software requires discharge estimates at the upper limit of the study site (cross-section 1, Figure 1). These estimates were calculated using the transect velocity–area method, based on velocity measurements collected throughout the February and March site visits [5,25].

Simulations were run for ice-covered and open-water conditions under two scenarios (lower water levels derived from ice-covered field data on 23 February 2021, and higher water levels derived from open-water field data on 2 March 2021) [5]. Each simulation yielded direct estimates of the shear velocity for each node along the lower boundary, which were used to calculate the shear stress along the channel bed using Equation (4):

$$\tau_b = \rho u_*^2, \tag{4}$$

where $\tau_b$ is the bed shear stress, $\rho$ is the water density, and $u_*$ is the shear velocity. When ice cover was present, the calculated shear velocity only applied to the channel bed and did not allow for the calculation of shear stress along the ice surface [22]. Model validation was conducted by calculating mean absolute error (MAE) values between simulated and measured velocities, ice surface elevations and water levels at calibration point locations [6]. MAE was calculated using Equation (5):

$$MAE = \frac{1}{N} \sum_J^N abs(M_j - P_j), \tag{5}$$

where N is the number of observations, $M_j$ is the modeled value, and $P_j$ is the measured value [33].

## 4. Results

### 4.1. Cross-Section Velocity Distribution

Cross-sectional geometry analyses and changes in velocity through each cross-section were achieved by creating a mesh grid with cross-section widths corresponding to x values, depths corresponding to y values, and streamwise velocities corresponding to z values. This grid was used to create cross-sectional maps of velocity to visualize the lateral distributions for each cross-section. The lateral and vertical velocities were represented using a vector plot overlain on the cross-sectional contour, demonstrating the direction and magnitude of secondary velocities [29]. For ice-covered velocities, the intervals between the measurement locations were omitted to avoid errors caused by interpolation. The largest changes in streamwise velocities were observed within the pool, especially along the left bank of cross-section 4, where upstream velocities exhibited a notable increase under ice and a shift in the secondary flow toward the right bank rather than the left (Figure 3). The flow structures observed in open water, such as the circulation seen in cross-section 5 (Figure 3) and the helix-shaped flow cell in cross-section 6 (Figure 3), were not readily observed under ice cover on 23 February, which was likely due to the sparse data availability under ice. The cross-sectional velocity analysis hints at potential changes in velocity distribution during ice cover in the pool (Figure 3), but because of the nature of data collection in the winter, it was difficult to generate meaningful results for riffle cross-sections (Supplementary Materials Figure S2).

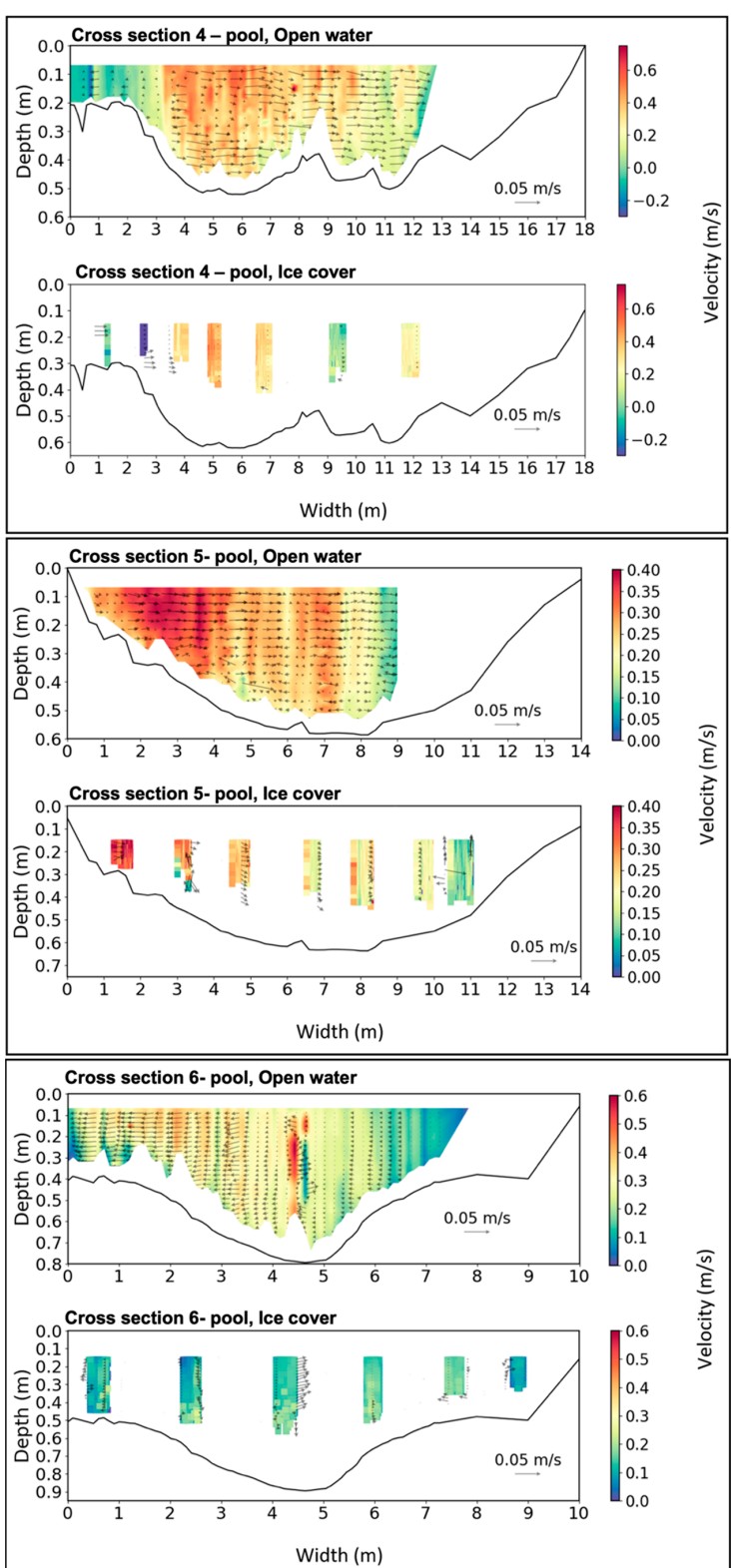

**Figure 3.** Velocity distribution maps along cross-section 4 (**upper panel**), 5 (**middle panel**) and 6 (**lower panel**) within the pool of the study site (Figure 1), from left bank to right bank under open-water conditions (top panel) and ice-covered conditions (bottom panel). Streamwise velocities are represented on a color scale and secondary velocities are represented by arrows showing direction and magnitude of the transverse and vertical velocity component. Data were collected using a Sontek S5 ADCP under ice-covered conditions on 23 February 2021, and open-water conditions on 2 March 2021.

### 4.2. Water Depth Distribution and Depth-Averaged Velocity

River2D modeled water depth under various conditions (Figure 4). In general, when ice cover was present, the water depth increased, with its maximum under higher flow, ice-covered conditions (Figure 4b), and at its minimum under lower flow, open-water conditions (Figure 4c). River2D does not allow for a fixed ice cover; thus, the water depths increased when ice was present to maintain flow continuity under a higher flow resistance. Outflow water depths were constant between the two lower flow simulations and the two higher flow simulations due to fixed water surface elevations as measured in the field (Figure 4).

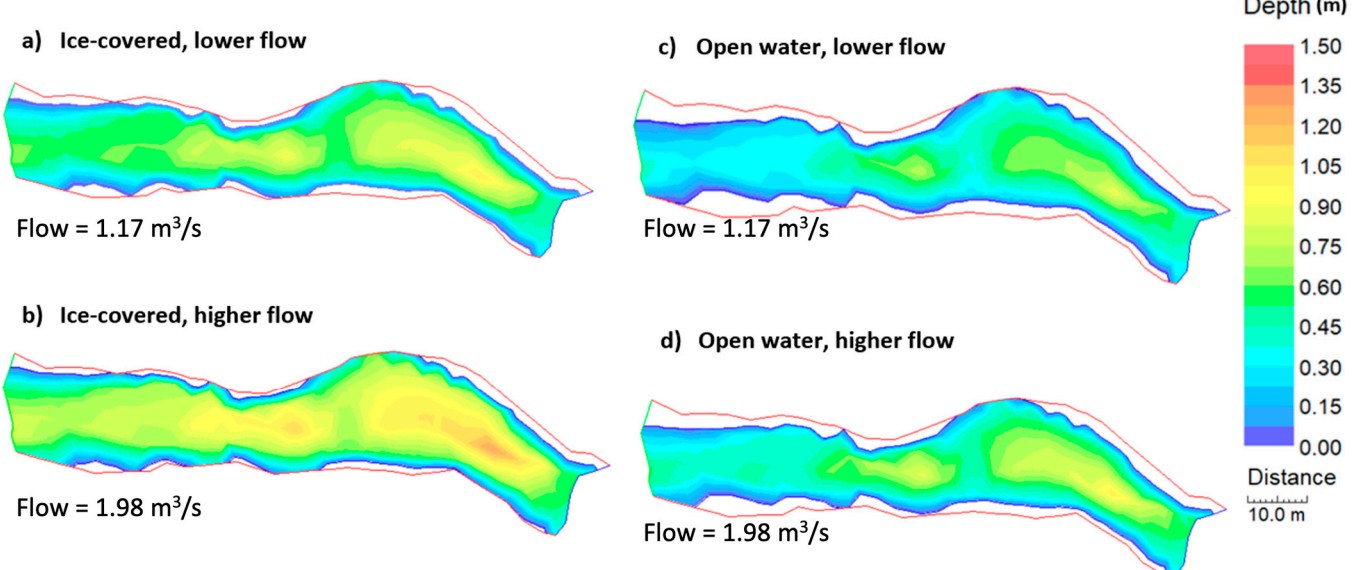

**Figure 4.** Modeled water depth under (**a**) lower flow, ice-covered conditions; (**c**) lower flow, open-water conditions; (**b**) higher flow, ice-covered conditions; and (**d**) higher flow, open-water conditions. Prepared in River2D.

Field data collection is challenging under ice cover; thus, models, such as River 2D, are useful tools to help characterize velocity flow patterns when ice is present. Simulations were run for ice-covered and open-water conditions based on lower flow and higher flow field data [5]. The lower flow, ice-covered and higher flow, open-water simulations were used for model calibration, validation purposes, and direct comparisons of depth-averaged velocities under different flow characteristics. No significant difference in the depth-averaged velocity magnitude was observed across the reach during lower flow conditions (Figure 5). However, depth-averaged velocities observed downstream of the pool directly above the outflow of the model under ice cover abruptly decreased and then increased (~0.4 m/s) in the River2D output (Figure 5a). There were slight changes to the wetted perimeter and thalweg width. Under lower flow, ice-covered conditions, the wetted perimeter increased (Figure 5a) when compared to the open-water, lower flow conditions (Figure 5c). The upstream thalweg width was largest under open-water conditions (Figure 5a vs. Figure 5c), indicating a wider spatial distribution of faster velocities. Along the left bank of the pool, a recirculating eddy was more obvious under ice-covered conditions compared to open-water conditions (e.g., Figure 5a vs. Figure 5c). Flow direction vectors were a direct output of River2D, and the depth-averaged flow directions were predominantly streamwise throughout the length of the channel, except for the recirculating eddy (Figure 5).

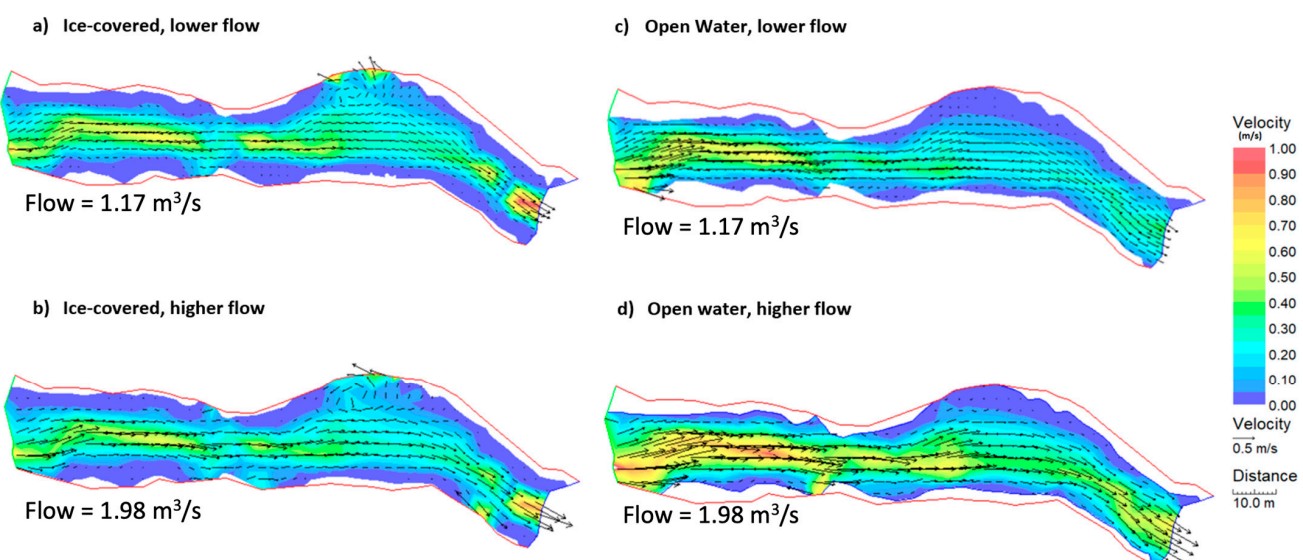

**Figure 5.** Modeled depth-averaged velocities during (**a**) ice cover, lower flow conditions (1.17 m$^3$/s); (**b**) ice cover, higher flow conditions (1.98 m$^3$/s); (**c**) open-water, lower flow conditions (1.17m$^3$/s); and (**d**) open water, higher flow (1.98 m$^3$/s). Flow directions are indicated at 1 m intervals using vector arrows (if they were plotted at 0.1 m intervals it would be difficult to visualize), and no-flow boundaries are outlined in red; the inflow boundary is on the left and outflow boundary on the right. Prepared in River2D.

Similar results were generated for higher flow conditions in open water and ice-covered flow (Figure 5b,d). The wetted perimeter increased under ice as demonstrated by an increase in the wetted area from bank to bank (Figure 5b vs. Figure 5d). The thalweg widened under open water, and a notable increase was observed in velocities throughout the upstream section and within the pool in open-water higher flow (Figure 5d). The depth-averaged velocity magnitude under higher flow open-water conditions ranged from 0.6 m/s to 1 m/s, while ice-covered higher flow conditions exhibited a slower thalweg velocity range between 0.4 m/s and 0.6 m/s (Figure 5b,d). As demonstrated under lower flow, ice-covered conditions (Figure 5a), an abrupt decrease followed by an increase in velocities was observed under higher flow, ice-covered conditions within the downstream, bottleneck section of the channel. This change exceeded velocities in the same location under open-water, higher flow conditions (Figure 5d). The eddy along the left bank of the pool was also present under higher flow conditions, and was exacerbated by the presence of ice cover.

The recirculating eddy along the left bank of the pool exhibited notable differences in velocities between open-water and ice-covered conditions for both lower and higher flow simulations (Figure 6). While still present under open-water conditions, the recirculating eddy along the left bank of the pool only exhibited velocities ranging between 0 and 0.1 m/s (Figure 6). Conversely, under ice-covered conditions, velocities within the eddy ranged from 0.1 to 0.6 m/s. Although higher flow open-water conditions did result in increased velocities when compared to ice-covered conditions, this increase in discharge under open-water conditions was concentrated within the thalweg across the pool, rather than in the eddy (Figure 6). Additionally, the eddy demonstrated slightly faster depth-averaged velocities under lower flow ice-covered conditions than under higher flow ice-covered conditions (Figure 6). However, velocities within the pool thalweg during higher flow, ice-covered conditions exceeded those seen in the same location but with a lower flow (Figure 6). The largest velocities within the pool thalweg occurred under higher flow, open-water conditions, ranging between 0.35 and 0.65 m/s (Figure 6).

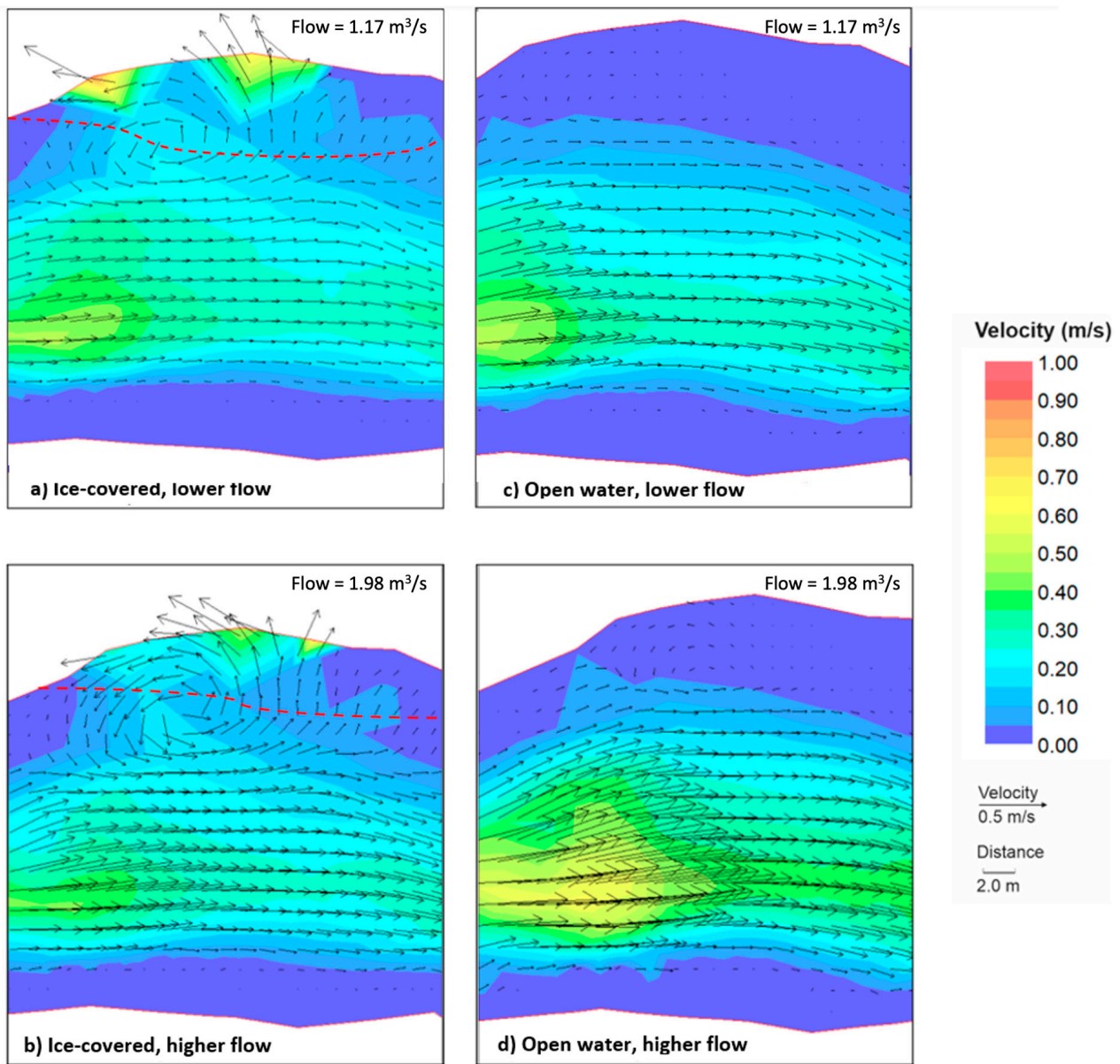

**Figure 6.** Depth-averaged velocity magnitude and directions within the pool under (**a**) lower flow, ice-covered conditions; (**c**) lower flow, open-water conditions; (**b**) higher flow, ice-covered conditions; and (**d**) higher flow, open-water conditions. Flow directions are indicated at 1 m intervals using vector arrows (if they were plotted at the mesh resolution of 0.1 m intervals it would be difficult to visualize). Ice extent along the left bank is delineated by dashed line in (**a**,**b**). Prepared in River2D.

The number of data points collected on 2 March 2021 during higher flow conditions without ice cover were more numerous than the data collected in February (with ice cover). Specifically, the ADCP collected data continuously throughout the site, as it was mounted to a float and towed throughout the reach. There were minimal differences in average velocities when the 2 March 2021 data were 'resampled' to data densities similar to the ice-covered collection. However, the velocity direction within the previously identified recirculation zone along the left bank of the pool was less obvious (Figure 7). These differences highlight the potential biases due to sample density during the open-water conditions.

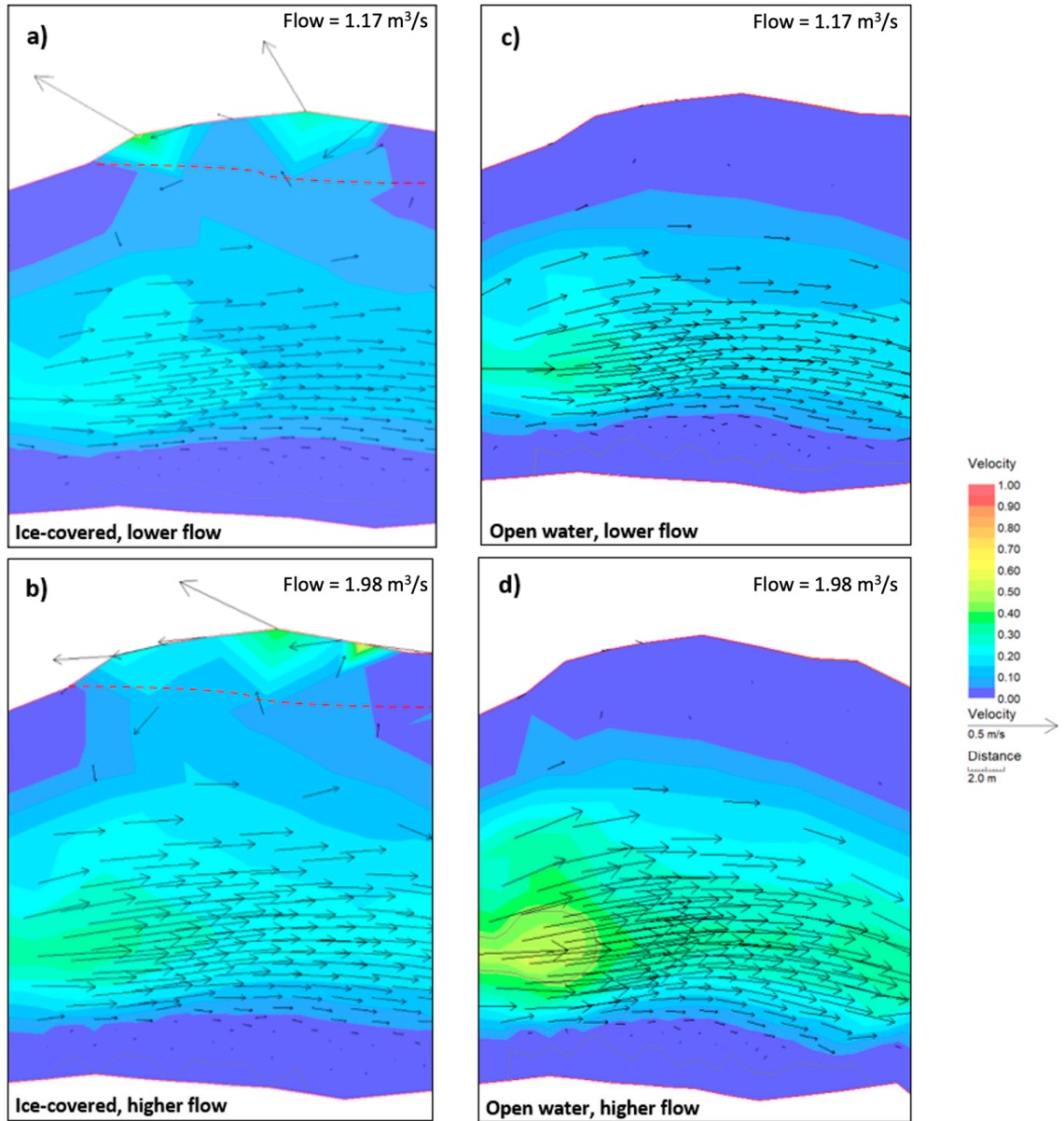

**Figure 7.** Resampled depth-averaged velocity magnitude and directions within the pool under (**a**) lower flow, ice-covered conditions; (**c**) lower flow, open-water conditions; (**b**) higher flow, ice-covered conditions; and (**d**) higher flow, open-water conditions. Flow directions are indicated using vector arrows at each mesh point, set at 1 m intervals. Ice extent along the left bank is delineated by dashed line in (**a**,**b**). Prepared in River2D.

### 4.3. Shear Stress Magnitude and Distribution

The shear stress values were calculated for all mesh nodes based on the River2D shear velocity output. Under all simulations, the upstream section of the river segment exhibited bed shear stress values exceeding $1\,\text{N/m}^2$ (Figure 8). Under open-water conditions, the shear stress hotspots spanned a wider area of the channel, while under ice-covered conditions, the hotspots narrowly follow the thalweg (Figure 8). The bed shear stress magnitudes were larger under higher flow conditions, ranging from 2 to $16\,\text{N/m}^2$ within the riffles, and from 0 to $1.5\,\text{N/m}^2$ within the pool (Figure 8b,d). Under ice-covered lower flow conditions, the bed shear stress values remained low throughout the pool, only increasing above $1\,\text{N/m}^2$ within the recirculating eddy along the left bank. The maximum bed shear stress values increased under lower flow ice-covered conditions in the upstream

and downstream sections, while they remained smaller within the pool when compared to lower flow open-water conditions (Figure 8a,b). Overall, the widest distribution of high bed shear stress values occurred under open-water, higher flow conditions, exceeding those seen in all other simulations (Figure 8d). Lower flow, open-water conditions experienced a similar distribution but at a lower magnitude (Figure 8c). When ice cover was present, regions exhibiting a high bed shear stress shifted, with hotspots observed along the left bank of the pool and upstream and downstream of the pool (Figure 8a,b). The most notable differences between the ice-covered and open-water shear stress distributions were noted along the recirculating eddies within the pool and the downstream section located above the outflow boundary (Figure 8). Overall, the range in the bed shear stress magnitude did not vary much between open-water and ice-covered conditions, but larger bed shear stress values were observed under higher flow conditions.

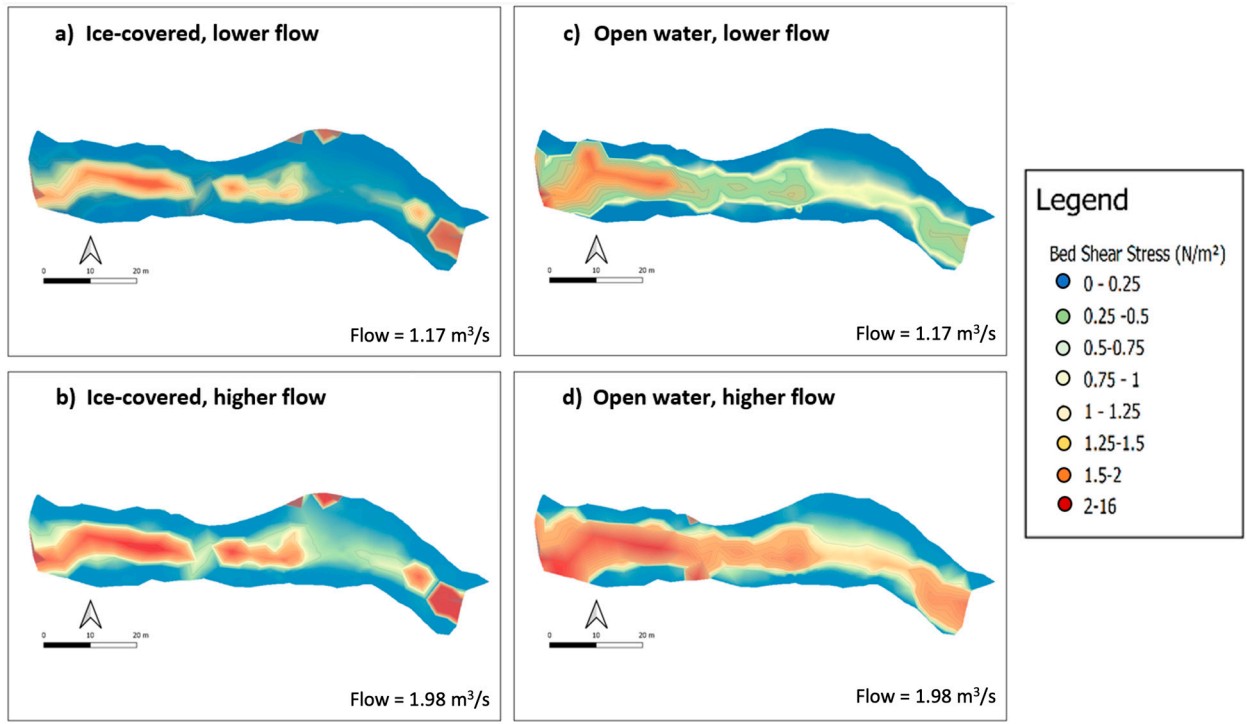

**Figure 8.** Bed shear stress distribution under lower flow, ice-covered conditions (**a**); lower flow, open-water conditions (**c**); higher flow, ice-covered conditions (**b**); and higher flow, open-water conditions (**d**). Prepared in QGIS.

Based on field bed and channel geometry surveys (Table 1), nodes within the modeled mesh were classified using the cross-sectional depth to median grain size ratio (d/D) to evaluate the shear stress magnitude variability as a function of the geomorphic unit (e.g., riffle vs. pool) (Figure 9) [34]. The shear stress magnitudes for pool cross-sections (d/D > 10) were lower than the shear stress for riffle cross-sections (d/D < 10) (Figure 9). The shear stress along pool cross-sections was less than 1 N/m², indicating low shear stress values, while the shear stress at riffle cross-sections ranged above 1 N/m² (Figure 9). *p*-values from a two-tailed t-test indicate significant differences in the bed shear stress values within a 95% confidence interval for the pool under higher flow conditions and the riffle under lower flow conditions (Table 2, Figure 9). However, the *p*-values for the riffles under higher flow conditions and the pool under lower flow conditions did not indicate significant differences between the shear stress magnitude under open-water and ice-covered conditions (Table 2, Figure 9).

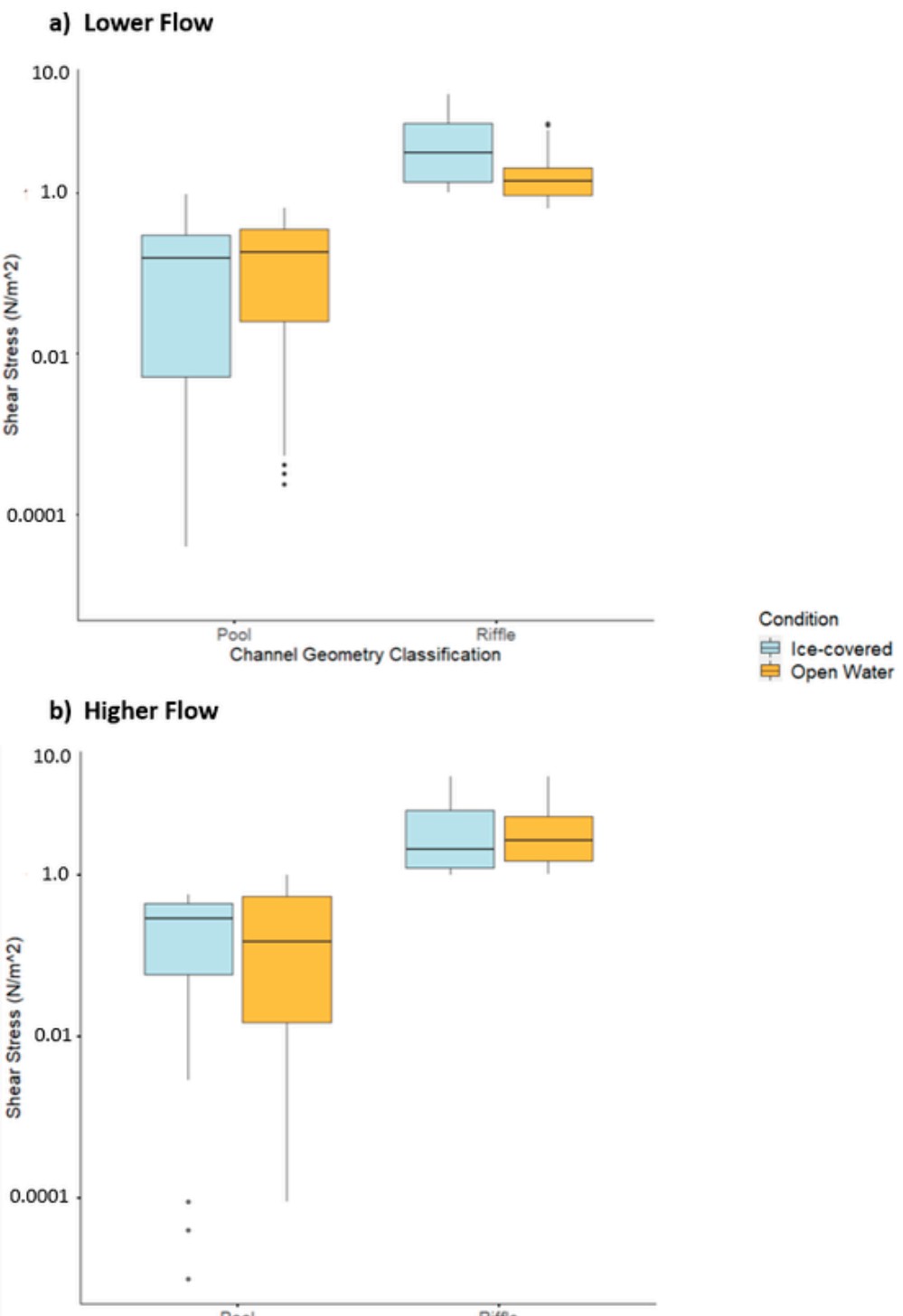

**Figure 9.** Bed shear stress distribution under (**a**) lower flow and (**b**) higher flow conditions based on the channel geometry classification of depth over median grain size (Table 1).

**Table 2.** *p*-values from difference of means test conducted between ice-covered and open-water conditions in the pool (d/D > 10) and riffle (d/D < 10) for lower and higher flow conditions.

|  | Lower Flow | Higher Flow |
|---|---|---|
| Pool (d/D > 10) | 0.26 | 0.041 |
| Riffle (d/D < 10) | <0.0001 | 0.053 |

### 4.4. Model Validation Results

A model validation demonstrated that the lower flow simulation results were moderately stronger than those from the higher flow (Table 3). No real-time observations were conducted for lower flow, open-water conditions or higher flow, ice-covered conditions; therefore, model validation was not calculated for those simulations. The lower flow model was most successful in simulating depth-averaged velocities, within 0.046 m/s, with water depths being simulated within 0.098 m and shear stress within 0.147 N/m$^2$ (Table 3). Under higher flow, open-water conditions, the depth-averaged velocities exhibited a marginally lower MAE, while the MAE for water depth was halved, and the MAE for shear stress was slightly larger (Table 3). Overall, the shear stress demonstrated the lowest success in modeled values for both lower flow and higher flow simulations, and depth-averaged velocities demonstrated the highest success (Table 3). Under the higher flow simulation, all average MAE values fell under 0.2, and the mean absolute percentage errors fell below 20% (Table 3). Additionally, the MAE values on the resampled datasets were conducted and follow the same patterns as the denser dataset analysis. The resampled MAE values also indicated a lower accuracy when compared to field values.

**Table 3.** Calculated MAE values for the modeled water depth, depth-averaged velocities, and shear stress values under lower flow, ice-covered conditions and measured data on 23 February 2021, and higher flow, open-water conditions and measured data on 2 March 2021. MAE values for the downsampled simulations models are shown in parentheses.

| Cross-Section | Ice-Covered Conditions, Feb 23 Field Data (Lower Flow) | | | Open-Water Conditions, Mar 2 Field Data (Higher Flow) | | |
|---|---|---|---|---|---|---|
| | Water Depth (m) | Depth-Averaged Velocity (m/s) | Shear Stress (N/m$^2$) | Water Depth (m) | Depth-Averaged Velocity (m/s$^2$) | Shear Stress (N/m$^2$) |
| 1 | 0.038 (0.12) | 0.018 (0.19) | 0.22 (0.36) | 0.035 (0.15) | 0.018 (0.13) | 0.25 (0.17) |
| 2 | 0.162 (0.22) | 0.065 (0.18) | 0.25 (0.27) | 0.098 (0.11) | 0.033 (0.16) | 0.13 (0.20) |
| 3 | 0.177 (0.24) | 0.079 (0.21) | 0.39 (0.19) | 0.063 (0.18) | 0.14 (0.15) | 0.36 (0.39) |
| 4 | 0.043 (0.17) | 0.036 (0.13) | 0.11 (0.23) | 0.031 (0.12) | 0.026 (0.09) | 0.073 (0.16) |
| 5 | 0.134 (0.23) | 0.058 (0.16) | 0.15 (0.18) | 0.012 (0.08) | 0.011 (0.10) | 0.11 (0.13) |
| 6 | 0.157 (0.28) | 0.077 (0.21) | 0.098 (0.21) | 0.014 (0.15) | 0.052 (0.14) | 0.093 (0.18) |
| 7 | 0.066 (0.15) | 0.016 (0.11) | 0.083 (0.14) | 0.091 (0.11) | 0.012 (0.09) | 0.047 (0.16) |
| Reach Average | 0.11 (0.20) | 0.050 (0.17) | 0.19 (0.23) | 0.047 (0.129) | 0.041 (0.123) | 0.152 (0.199) |

Model validation demonstrates that the model output accuracy varied between the criteria and location throughout (Table 3). Pool cross-sections exhibited lower MAE values, while riffle cross-sections exhibited higher MAE values (Table 3). These results indicate that the River2D-simulated ice-covered flow in this reach of the Sixteen Mile Creek is sensitive to water depth, and that shallower flows are more problematic. River2D employs the St. Vernant equation to calculate depth-averaged velocities by considering the conservation of mass and momentum in the x and y direction [22,30]. Studies have demonstrated that in shallow, turbulent waters, with significant secondary flow components, the accuracy of this equation may be affected [35]. Furthermore, locations with shallower, slower-moving waters will be proportionally more sensitive to mean absolute error values.

## 5. Discussion

Higher and lower flow conditions largely influence velocities, flow directions, and shear stress in channels. There are limited data on the interaction between discharge and ice cover as it relates to fluvial processes in small channels [36,37]. The lower and higher flow scenarios tested in this study were representative of a range of conditions experienced

in winter 2021. The nearest Water Survey of Canada (WSC) gauging station is 4.5 km upstream of the study site (Station ID 02HB005); from this record, the winter 2021 average discharge was 1.2 m$^3$/s (long-term winter season average was 1.4 m$^3$/s, 65 years) [23]. At the WSC gauging station, the discharge was 0.625 m$^3$/s on 23 February 2021 and 1.22 m$^3$/s on 2 March 2021 [23]. These values are slightly lower than the conditions measured at the study site and are reflective of the upstream position of the WSC gauge. However, given the relative differences between our study site and the WSC data, we can conclude that the February 23 field data coincided with lower flow conditions, likely below the seasonal winter average, and the March 2 field observations coincided with average winter flow conditions but represent the higher conditions of the two comparison datasets.

### 5.1. Spatial Distribution of Depth-Averaged Velocities and Bed Shear Stress under Ice Cover

Under both higher and lower flow conditions, the outflow water surface elevation was constant between ice-covered and open-water trials, therefore reasonably simulating a fixed ice cover (Figure 4; Supplementary Material Figure S3). Calibrated model-simulated depth-averaged velocities and water levels were within a 10% margin of error for both simulations (Table 3). Experimental flume studies demonstrated that a fixed ice cover leads to confined flow and increased depth-averaged velocities [12], which were similar to the observed values downstream of the pool in this study (Figure 5). However, water depth within the pool and in the upstream riffle increased under ice cover, indicating that incorporating ice cover in River2D does not explicitly consider bed- or bank-fast ice unless the water depth is maintained constant, as seen near the outflow (Figure 5) [38]. Throughout winter 2021, the ice cover did not fully extend across the pool, leaving a strip of open water along the left bank and preventing fixed ice and confined flow from occurring in the pool (Figure 2). The results from the velocity simulations demonstrated overall slower depth-averaged velocities under ice cover when compared to open water for both the lower flow and higher flow outputs (Figure 5). However, the recirculating eddies along the left bank of the pool had higher velocities when ice was present, and the segment downstream of the pool presented exceptions to these results (Figure 5). In pool-riffle sequences, increases in velocity downstream of pools occur to maintain flow continuity following channel narrowing [38]. Peters et al. [20] found that when ice cover was not fixed to the bed or banks, increases in the flow resistance from the upper boundary increased the water depths and reduced the bulk velocities. The changes in depth-averaged velocities in higher flow, open-water conditions as compared to higher flow, ice-covered conditions were more notable than under lower flow conditions, suggesting that in these simulations, the ice cover impact on velocities is exacerbated under higher flow conditions (Figure 5).

The thalweg location remained relatively consistent in all simulations, although the thalweg was distinctly wider under open-water conditions (Figure 5c,d). The 'sandwiching' effect that the ice, bank, and bed have on velocities prevents faster velocities from being located close to the banks, which effectively narrows the thalweg in the channel when ice cover is present [39,40]. Previous studies found changes in the thalweg location under ice cover conditions, noting that faster flows were concentrated in narrower areas [6,11]. However, in this study, there was limited evidence from the measured data (Figure 4) to suggest a concentration of the thalweg when ice was present. Overall, the simulated spatial distribution of velocities under ice remained constant under both flow conditions (Figure 5). However, recirculation and upstream flow were more prevalent under higher flow conditions, particularly when ice cover was present (Figures 5 and 6). This indicates that although changes in discharge may not significantly impact the distribution of slower and faster velocities under ice, they can enhance recirculation and cause faster bulk velocities. Turbulent flow in riffle-pool sequences typically increases in prevalence and magnitude under higher discharges, and this was observed between lower and higher flow conditions, which was further exacerbated by the presence of ice [38].

Bed roughness did not have a consistent impact on depth-averaged velocities under ice cover (Figure 5a,c). Field observations at the same location [5] suggest no reduction

and, in some cases, slight increases in the maximum velocities under ice at cross-sections within riffles, while cross-sections exhibiting lower roughness (e.g., in the pool) noted a decrease in maximum velocity magnitudes (Figure 4). Conversely, model simulations only noted increases in depth-averaged velocities under ice along the riffle cross-section downstream of the pool (cross-section 7 (Figure 1), Figure 5a,b). These results demonstrate that while ice-covered depth-averaged velocities may be reduced in magnitude along riffle cross-sections, maximum velocities may not show the same trend [13,41]. Furthermore, as River2D is able to simulate depth-averaged velocities, the ability to interrogate changes in maximum velocities within the water column is limited.

Another notable change in the depth-averaged velocity magnitude under ice cover was observed along the left bank of the pool (Figure 6). In each model output, a recirculating eddy was present along the left bank edge of the pool (Figures 5 and 6). Vertical and horizontal eddies commonly occur in pools located directly downstream of fast-flowing riffles as a result of sudden changes in the channel geometry [42]. However, during both lower and higher flow simulations, the eddy exhibited faster depth-averaged velocities when ice cover was present (Figure 6). This is supported by the velocity mapping work for each of the data collection dates (Figure 4, upper panels in a–c) and from velocity magnitude and direction mapping across each cross-section in the field [5]. The location of the eddy coincided with the strip of open water located along the left bank during ice-covered conditions (Figures 1 and 2) [5]. Partial ice cover can cause a redistribution of depth-averaged velocities, resulting in slower velocities directly under the ice cover and displacing faster flow to uncovered (unconfined) locations within the channel [20]. The velocities within the pool were reduced under ice cover due to increases in resistance, and faster velocities were redistributed from the thalweg toward the ice-free left bank, therefore intensifying the eddy (Figure 6). An experimental study by Peters et al. [20] had similar findings, where faster velocities were located under open-water segments of a partially covered flume.

Bed shear stress is dependent on multiple variables, including shear velocity magnitude, bed roughness, and the relative depth of maximum velocity within the water column [43]. While the presence of ice has no immediate impact on bed roughness, the upper and lower boundaries' interacting influence on depth-averaged velocities can influence velocity gradients and, therefore, shear velocities [4]. This causes different impacts on the bed shear stress under the ice within riffles and pools [7]. The results from the simulations demonstrated that under both lower and higher flow, open-water conditions exhibited a wider distribution of bed shear stress values exceeding $0.5 \, \text{N/m}^2$, following closely along the thalweg (Figure 8). These results mirror the narrowing in thalweg observed under ice-covered conditions, which is similar to other studies that confirmed that bed shear stress values are dependent on the thalweg location [44]. The bed shear stress did not exceed $16 \, \text{N/m}^2$ in any simulation or scenario and is consistent with shear stress values observed in small channels exhibiting riffle-pool sequences [5,24,45]. The higher flow simulations demonstrated the similar maximum and minimum values as lower flow conditions, although bed shear stress values exceeding $1 \, \text{N/m}^2$ covered a larger area under the higher flow conditions (Figure 8). This suggests a wider distribution of large bed shear stress values under higher flow conditions (Figure 8). Bed shear stress is known to increase with higher velocities, therefore increasing the shear stress under higher flow conditions [18]. For both higher and lower flow conditions, the results from the bed shear stress analysis indicated that when ice cover was present, the shear stress increased in the riffle directly upstream of the pool (cross-section 3, Figure 1), as well as downstream of the pool (Figure 8). Additionally, the bed shear stress was greater along the left bank of the partially covered pool in the same location as the recirculating eddy, which was caused by faster depth-average velocities (Figure 8).

Changes in depth-averaged velocities alone do not explain the changes in the shear stress observed through these simulations (Figure 8). Previous work has demonstrated that ice cover can push the maximum velocities closer to the bed, increasing the velocity

gradient and resulting in faster shear velocities [5,18,25]. Field studies at Sixteen Mile Creek demonstrated that maximum velocities were located closer to the bed under ice-covered conditions and coupled with negligible changes in magnitudes of maximum velocities, the bed shear stress was found to increase both upstream and downstream of the pool cross-sections [5]. Conversely, in cross-sections with a lower level of roughness within the pool, maximum velocities were notably reduced and did not result in an increase in shear stress under ice [5]. Modeled shear stress values were not consistently different between ice cover and ice-free conditions (Figure 9; Table 2). Similar to the field results [5] and previously published studies including flume experiments [13,18], the bed shear stress was larger in channels with greater roughness (e.g., riffles) than in lower roughness channel segments (e.g., pools). River2D does not take into consideration the depth of maximum velocities when calculating the shear velocities [22]. Additionally, the shear velocity output is derived using the composite roughness of the upper and lower boundaries and it may not present an accurate picture of the shear velocities along the bed under the ice [6]. This could explain why the modeled shear stress values within rough and smooth cross-sections under ice and open-water conditions do not agree with the findings from the field [5]. However, while the shear stress magnitude is not as accurately calculated by the model, the simulations provide detailed insight into the distribution of shear stress. This enables the identification of high and low shear stress zones and demonstrates changes caused by ice cover. Ice-impacted river management can benefit from understanding where zones of high shear stress shift when ice cover is present, therefore informing seasonal erosion and deposition processes.

*5.2. Implications for Ecosystem Services*

There is a known population of silver shiners, an endangered minnow species, that utilizes the left bank of the pool during open-water conditions [46]. Their preferred habitat includes riffle-pool sequences, which are generally wider than 30 m, although they are occasionally found in smaller tributaries such as Sixteen Mile Creek [47]. Previous work on silver shiners reports that they are commonly found within the water column where velocities are ~0.12 m/s ($\pm$0.02 m/s; [47]). The velocities along the left bank at this site ranged between 0.01 and 0.18 m/s in open water but exceeded 0.40 m/s when ice partially covered the pool (Figure 6), potentially reducing the usable silver shiner habitat during the winter season. Furthermore, previous works (e.g., Huusko et al. [48]) have demonstrated the need for flow variability such as this, to offer refuge, but also, to offer a steady supply of materials from upstream. The recirculation zone along the left bank offers a point for fish to station hold while materials move downstream closer to the middle of the channel, acting as an almost food conveyor belt.

Increased velocities along the eddy in the partially covered pool may present insight into pool maintenance mechanisms during ice-covered conditions. Riffle-pool sequences are common geomorphic units in low- to moderate- gradient channels and are important influences on local hydraulics, sediment transport, and aquatic habitat [49,50]. The periodic removal of fine sediment deposited along pools is essential for maintaining pool morphological characteristics [51]. Geomorphologists have presented several potential mechanisms for riffle-pool maintenance, including velocity reversal, stormflow, flow convergence, and natural hydrogeomorphic maintenance [38,49,52,53]. Within the pool, the presence of partial ice cover increased the depth-averaged velocities along the left bank (Figure 6). This suggests a potential increase in sediment transport and the removal of fine-grained sediment from the pool under both lower and higher flow conditions, especially those with a partial ice cover. The grain size along the left bank of the pool was dominated by sand and silt-sized material (Table 1), and the critical velocity required for entrainment was ~0.19 m/s [54]. While open-water depth-averaged velocities may be insufficient for entrainment, velocities along the left bank when ice cover was present exceeded the critical velocity of 0.19 m/s under both lower and higher flow conditions (Figure 6). If consistent over a long period of time, the impact of ice cover on pool depth-averaged velocities could play a

role in pool maintenance in ice-affected riffle-pool systems. Additionally, the recirculation zone found along the pool's left bank may provide an important ecosystem function for silver shiners by transporting nutrients and food sources into the channel [55].

*5.3. Modeled Shear Stress Distribution and Magnitude: Advantages and Drawbacks*

Simulated shear stress during ice presence in the channel reached up to 2 N/m$^2$, suggesting variability in the erosional potential between ice-covered and open-water conditions (Figure 8). A similar study by Lotsari et al. [6] investigated spatial variations in depth-averaged velocities under ice along a Pulmanki River meander bend. The results from the Pulmanki River study indicated that near-bed velocities during all seasons would be sufficient for incipient motion, but that the spatial variability in erosion–deposition locations increased under ice [6]. Lotsari et al. [6] stated that ice cover should be considered as an important factor in morphological changes to meander bends, and the findings from the Sixteen Mile Creek suggest that the same is true for riffle-pool sequences [5]. Moreover, the value in mapping shear stress across the channel using River2D highlights the opportunity to see changes in the shear stress distribution across the bed, which can be limited only when the field data are evaluated (e.g., Smith et al. [5]). High-resolution velocity data collection (e.g., Figure 3) is challenging with ice cover present, as the model facilitates an additional opportunity to interrogate the data.

The bed shear stress estimates exhibited higher error values than depth-averaged velocities and water levels (within a 30% margin of error) (Table 3). There are several factors that influence the accuracy of shear stress estimates. The field bed shear stress estimates were calculated using logarithmic velocity profiles to identify velocity gradients and calculate shear velocity and shear stress [4,5]. However, River2D calculates shear velocity using the Keulegan equation, which incorporates the mean velocity, water depth, and equivalent sand roughness ($k_s$) [22,56]. Methods that use different variables to calculate shear stress can lead to varying results and increase the margin of error between field-based and simulated estimations [24]. The logarithmic (law of the wall) method does not consider antecedents along the bed or acceleration and deceleration within riffle-pool transitions, while the Keulegan equation simplifies the influence of bed roughness using an equivalent roughness height coefficient [57,58]. Ice-covered simulated bed shear stress values may be further influenced by the composite roughness value used in River2D to calculate shear velocities, leading to a reduced accuracy [6]. Shear velocity model outputs can be used conservatively to estimate bed shear stress but should be coupled with field surveys to ensure accuracy in bed shear stress magnitude estimates.

## 6. Conclusions

The River2D hydrodynamic model successfully simulates flow characteristics in rivers of varying sizes and can be used for applications, such as habitat suitability models, river restoration plans, and sediment transport estimates [6,9]. This research simulated velocity distribution and shear stress in a small, shallow riffle-pool sequence under ice cover and through two different flow levels. The results indicated reduced depth-averaged velocities throughout the upstream riffle and pool, and increased depth-averaged velocities along the left bank and downstream of the pool under ice cover for both higher and lower flow simulations. The thalweg widened under ice cover, and the spatial distribution of the velocities shifted when ice cover was present, exhibiting a higher amount of recirculating flow along the partial ice cover in the pool. Increased discharge resulted in larger differences in water depth and velocities between open and ice-covered conditions, although the spatial distribution of flow remained consistent between the two ice-covered and open-water simulations. The recirculating eddy intensified with the presence of ice, increasing the depth-averaged velocities in known silver shiner habitats. Additionally, the findings from velocities within the pool suggest that ice cover may play a role in pool maintenance in ice-affected streams.

The shear stress analysis demonstrated no difference in the maximum and minimum shear stress values between open-water and ice-covered flow. However, there were clear differences in the distribution of bed shear stress under ice and open water for lower and higher flows. Overall, the model was successful in simulating depth-averaged velocities and water levels throughout the study site, while the shear stress values exhibited a higher and more variable margin of error. The shear stress values derived from River2D for small, ice-covered streams established a baseline approximation, but should be paired with field measurements for a more accurate picture. Ultimately, River2D was successfully calibrated to simulate depth-averaged velocities, flow directions, and water levels in a small and shallow riffle-pool sequence but was proven to be more effective in deeper sections.

**Supplementary Materials:** The following supporting information can be downloaded at: https://www.mdpi.com/article/10.3390/w15081604/s1. Figure S1: Problem schematic. Figure S2: Velocity distribution maps for each riffle cross-section (a is cross-section 1; b is cross-section 2; c is cross-section 3; d is cross-section 7; Figure 1) from left bank to right bank under open water conditions (top panel) and ice-covered conditions (bottom panel). Streamwise velocities are represented on a colour scale and secondary velocities are represented by arrows showing the direction and magnitude of the transverse and vertical velocity component. Data were collected using a Sontek S5 ADCP under ice-covered conditions on 23 February 2021, and open water conditions on 2 March 2021. Figure S3: Modelled water surface elevations for all four simulations. Water elevation decreases in the downstream direction according to the channel bed slope, and to simulated water depths.

**Author Contributions:** Conceptualization, K.S., J.M.H.C. and P.V.V.; methodology, K.S., J.M.H.C. and P.V.V.; data collection, K.S. and J.M.H.C.; formal analysis, K.S., P.V.V. and J.M.H.C.; writing—original draft preparation, K.S. and J.M.H.C.; writing—review and editing, J.M.H.C., K.S. and P.V.V.; supervision, J.M.H.C.; project administration, J.M.H.C.; funding acquisition, K.S., J.M.H.C. and P.V.V. All authors have read and agreed to the published version of the manuscript. **Funding:** This research was funded by the Canadian Foundation for Innovation, grant number: 31341. K.S. received additional stipends from the Yukon Foundation: Dorreene and Herb Wahl Award and Yukon Order of Pioneers Yukon Grant. The APC was waived.

**Data Availability Statement:** This article is based on the MSc. Thesis research was conducted by K.S. and available https://hdl.handle.net/10214/26885. Some data are proprietary, please contact the corresponding author for details.

**Acknowledgments:** We express our sincere gratitude to Abby Smith for epic field support on the coldest and longest of days! Further, field and technical support by GEO Morphix staff, specifically, Patrick Padovan, Tye Rusnak, Lindsay Davis, Marie Hoekstra, and Bryce Molder is greatly appreciated. Marie Puddister and Skyler Barclay in the Department of Geography, Environment and Geomatics at University of Guelph, provided production support. Additionally, we offer our sincere thanks to Mattamy Homes for the site access to Sixteen Mile Creek.

**Conflicts of Interest:** The authors declare no conflict of interest. The funders had no role in the design of the study; in the collection, analyses, or interpretation of data; in the writing of the manuscript; or in the decision to publish the results.

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
