# Peer review of "Rivers under Ice: Evaluating Simulated Morphodynamics through a Riffle-Pool Sequence"

_water, doi:10.3390/w15081604_

Round 1
Reviewer 1 Report
The authors simulated ice-covered velocities and shear stress under various flow conditions, and to assess model performance in a small, ice-covered riffle-pool sequence. This paper can be published in "Water" after major revisions:
1) A schematic of the problem should be presented.
2) The boundary conditions should be provided.
3) A validation case is required.
4) Grid study is required for the simulations.
5) I believe the flow is turbulence. Is is correct? Did you use a turbulence model?
6) The governing equations should be provided.
7) You should present the results to demonstrate the thermal characteristics.
Author Response
Thank you for the thoughtful suggestions to help us improve the manuscript. We are responding to the comments as numbered by the reviewer:
1 – We created and added a problem schematic into the Supplementary Material (Supplementary Materials 1).
2 – Boundary conditions are listed in our newly named section 3.0 Model Validation (formerly sections 2.2. And as suggested by another reviewer, input flow was added to all figures.
3 –River2D and MAEs were successfully used in similar studies (e.g., Lotsari et al 2019). We follow these examples in this study.
4 – A triangular irregular network (TIN) was used (as described in the second paragraph of section 3 (Pre-processing, Data Analysis & Model Validation, section was originally section 2.2)). As described in 3 above, following the examples of others we did not complete a grid study (e.g., Lotsari et al 2019).
5 – The flow is turbulent in this stream channel. River2D (https://www.river2d.ca/) handles these conditions.
6 – Governing equations are given as needed (e.g., equations 1-5).
7 – While thermal characteristics can be important, they were not needed in this study. The small differences in temperature between data collection days (Feb 23 and March 2, 2021) were minimal, and considered insignificant compared to the other variables observed in the field.
Reviewer 2 Report
General comments
The authors have conducted and described field research and mathematical modeling of the process of open channel flow under ice conditions. The work is particularly interesting from the point of view of river engineering and eco-hydrology. I recommend this paper for publication after minor revision.
Principal criteria
Scientific significance (good). The manuscript represents a contribution to scientific progress within the scope of Hydrology (however, the conclusions are characterized by some uncertainty due to the relatively small number of measurements).
Scientific quality (good). The results were discussed in an appropriate way, considering related works and including appropriate references.
Presentation quality (good). The scientific results and conclusions are presented in a clear, concise, and well-structured way.
Specific comments
The authors, in their review of the literature (Introduction section), forgot to mention another significant model of ice-water flow, although this one was developed in Canada where they work. The ice phenomena can be also investigated by applying the DynaRICE two-dimensional river ice dynamic model. The model was developed at Clarkson University [A] and applied to simulate rivers [B] and coastal lakes [C] under ice conditions.
[A] Shen, H.T. Mathematical modeling of river ice processes. Cold Regions Science and Technology 2010, 62, 3–13.
[B] Kolerski, T.; Shen, H.T. Possible effects of the 1984 St. Clair River ice jam on bed changes. Canadian Journal of Civil Engineering 2015, 42, 696–703.
[C] Kolerski T., Zima P., Szydłowski M. Mathematical Modeling of Ice Thrusting on the Shore of the Vistula Lagoon (Baltic Sea) and the Proposed Artificial Island, Water 2019, 11(11), 2297.
Technical Comments
Line 103 - Please check the values and units in table 1. I am not sure a Median Grain Size column.
Line 211 - wrong symbol for bed shear stress
Line 388-391 - Please add the units to the table 3
Author Response
Thank you for the thoughtful suggestions to help us improve the manuscript. Below we outline our response to the Specific and Technical Comments from the reviewer:
Specific Comments
We have incorporated Shen 2010 [line 34] into the introduction section when listing various models. DynaRICE is an important model that helps us to understand ice formation, transfer and break-up. We’ve focused our work around River2D as it has been applied in similar studies to ours (Lotsari et al 2019 [6]).
Technical Comments
Table 1 units and values. The units for each value in the table are included in each cell (e.g., cross-section width is 12 m at cross-section 1). With respect to median grain size (or D50), grain size is measured in millimetres. We’ve added D50 to the column title.
Symbol for bed shear stress was corrected.
Units were added to the column title in Table 3.
Reviewer 3 Report
This presents a case study with ice-covered conditions using a 2D model River2D. In general, the paper focused more on results analysis but less on modelling itself, which seems not consistent to the title. The detailed comments and suggestions are listed as follows:
1. For readers who want to know how to model the flow with ice-covered conditions, this paper can’t provide much valuable information. Two solutions are available: (1) Remove “Modelling rivers under ice” in the title; (2) Put more details on modelling parts, such as governing equations, ice treatment, numerical method, etc.
2. Table 1 shows the median grain sizes in different reaches, varying from coarse gravel to boulders. According to Fig.2, the bed slope is about 1%. Based on the above information, the study reach seems a mountainous creek, where bed load transport may be active.
a. Please provide more information regarding the bed slope and sediment transport information
b. Current modeling results have no influences from sediment transport. I wonder if bedload transport has influences on the flow results.
3. This study analyzed the water depth, velocity and shear stress. Another useful important parameter may be Froude number. If possible, please provide information of the Froude number distribution, so that the flow mode for the study reach can be identified.
4. Please check the reference “[21] Steffler, P.; Blackburn, J. Two-Dimensional Depth Averaged Model of River Hydrodynamics and Fish Habitat 2002.”, which can’t be found.
Author Response
Thank you for taking the time to provide helpful and clear suggestions for improving our paper. Below we outline our response to each comment/suggestion.
1. As suggested, we have revised our title “Rivers under ice: evaluating simulated morphodynamics through a riffle-pool sequence” to reflect that the paper does not develop a model, but rather demonstrates that the model used enables deeper understanding of these dynamics.
2. Bed slopes were added to the text (lines 133-135) in the second paragraph in section 2, and as noted during field visits sediment transport was minimal. Some suspended transport was noted on underwater cameras in the field study (Smith et al 2023, Smith 2022), and bedload transport is negligible, thus sediment transport considerations in the model were not needed (this note was added at the end of the first paragraph in section 2 (line 111-112).
3. The flow in this channel is subcritical (Froude number less than 1).
4. Reference information was corrected for Steffler & Blackburn 2002 [now ref 22]. We had it classified incorrectly in our reference manager software, here is the URL if the reviewer is interested - https://relicensing.pcwa.net/documents/Library/PCWA-L%20452.pdf
Reviewer 4 Report
The main observations are listed below. The acceptance of the manuscript would depend on the revision. The author needs to provide a point by point response or provide a rebuttal.
1. The abstract should be briefly written to describe the purpose of the research, the principal results, and major findings. Authors should revise it.
• one sentence to present the significance of the study,
• one sentence to present the aim of the study,
• one sentence to present the research methodology, and
• two sentences to present the conclusion drawn from the study.
2. In the Introduction, the literature review was not logically organized and all literatures cited seem separate descriptions without connections. The readers can’t know what the state-of-art methodologies or gaps the current study plans to resolve or fill, and how significant or what contribution the current study is?
3. The result and discussion section is the main weakness of your study and needs to be improved a lot. Physical justification behind the results must be provided for each and every graph. Authors have mentioned only the increasing/decreasing trend of the curves.
4. Authors should briefly explain how can hydrodynamic modelling provide valuable insight into seasonal changes in velocities and shear stress?
5. The conclusion part is weak. It's hard for readers to get useful information about this study?
6.What are the advantages of employed technique?
7. The Methodology and Validation are not clear. The author should divide them in two separate sections (Methodology -section and Validation -section).
Author Response
Thank you for taking the time to provide suggestions for improving our paper. Several of the points made by the reviewer were incorporated into our revised submission and/or were related to suggestions from other reviews. However, there were points that we disagreed with and have outlined our response to each comment/suggestion as numbered by the Reviewer.
1. Abstract was revised, but we disagree with the reviewer’s strict outline, and have written our abstract to best reflect the work and our perspective.
2 & 5. Minor edits were made throughout the paper, based on specific suggestions made by other reviewers. The manuscript has gone through several internal rounds of review at our institution, without this being an issue at any point.
3. Results and discussion sections were revised in conjunction with specific suggestions made by other reviewers. Additionally, the methods and model validation sections were separated, which we believe helps strengthen the paper overall.
4 & 6. This is described in the introduction (e.g., lines 28-33, & 80-81). Specifically, River2D is able to incorporate ice cover while evaluating velocity magnitude & distribution across the channel. Previous work (e.g., Lostari et al 2019) focused changes around a meander (and found significant changes to thalweg shape and distribution), in our study we investigated the bed morphology (e.g., riffle or a pool) impacts through high and lower flow conditions, with and without ice cover.
5. See response 2.
6. See response 4
7. Revised as suggested (see response 3).
Reviewer 5 Report
I want to thank the authors for their interesting paper.
The topic is important and the actual literature lacking on these aspects.
In general, the paper is well organised.
Nevertheless, I have some minor and major recommendations to suggest to the authors.
1. Sincerely, I don't understand why the water depth considering ice cover are higher than free surface flow. Please, better explain this point
2. lines 72-74. Not clear: You used field data during ice cover to calibrate the ice-free simulations? Why?
3. line 82. From figure 1b doesn't seems that the narrowest section is only 1 m
4. section 2.2. You never talk about the mesh resolution, how you decided it, if you perform a mesh resolution analysis and so on
5. 164. In this condition, ice presence, is the hydrostatic pressure distribution hypothesis still correct?
6. Figure 4. In my opinion it's better that you report for each graph, the flow rates used
7. Figure 5. From these figures, if the velocity vectors are one for each computational cell, it seems that the mesh resolution is about 1m. I think that it is too coarse. Please explain it and consider to increase the resolution in case.
8. Figure 6 a,b. These velocities values seems very strange. The velocity should reduce and goes to zero near the boundaries. Here you have a maximum of the velocities. Please better explain and in case correct your results.
9. Section 4.2. For me this section, even if interesting, can be removed.
Author Response
Thank you for taking the time to provide helpful and clear suggestions for improving our paper. Below we outline our response to each comment/suggestion as numbered by the reviewer.
1. Ice cover is not fixed in River2D, to maintain flow continuity (addressing increased resistance in ice cover) depth increases. While the range in velocity magnitude did not differ much between open water and ice-covered simulations, the distribution of high velocities is more restricted under ice-cover. This is consistent with the concept of flow continuity, where increased flow resistance will result in increased water depth and decreased velocities to maintain consistent discharge.
2. As requested by a different reviewer, we’ve added a schematic of the problem (see Supplementary Material Figure 1). Collecting velocity and depth data from a channel with ice cover is challenging, thus we wanted to see how well River2D could model velocity (and bed shear stress) under ice conditions while flow was ice-covered. As part of a field study, data was collected at the site under ice-covered, lower flow conditions in February 2021, and under open water, higher flow conditions in March 2021. Through modelling, we aimed to understand how velocity and shear stress would change in various flow conditions. Therefore, the field data was used to calibrate two control simulations which allowed us to validate River2D’s accuracy in representing ice-covered and open water hydrodynamics at Sixteen Mile Creek. This allowed for high confidence in the results for open water, lower flow and ice-covered, higher flow simulations, and in the comparison of velocity distributions across all four simulations.
3. Perhaps a misunderstanding here. Fig 1b is referenced in line 82 to show that the channel is ~20 m at its widest. The next part of the sentence (~1 m) is related to channel depth. Yes, as the reviewer states, the channel is more than 1 m wide.
4. Section 2.2 has now been renamed as 3. Pre-processing, Data Analysis & Model Validation as suggested by another review and the 10 cm mesh resolution information was added (206-207 & 208-211). Mesh resolution was set at 10 cm to provide sufficient details on hydrodynamics in the chosen site while still allowing for accurate validation between field data and modelled velocities for the control simulations. An ADCP was used to collect velocity data, and under ice cover, the measurement bins ranged from 2 cm – 10 cm in size.
5. Yes, the hydrostatic pressure distribution hypothesis is still valid, in River2D, ice cover is not fixed to the bed/bank of a channel. Moreover, there were small open sections (ice-free) along the left bank (Figure 1c) throughout the data collection process.
6. Flow rates were added to all figures, including Figure 4 as suggested.
7. The mesh resolution is 10 cm, but in order to effectively visualize velocity vectors, vectors were plotted at 1 m intervals. Text was added to figure captions (Fig 5 & 6) to explain the data visualization decision. Furthermore, mesh resolution for Figure 7, where data is downsampled, was set at 1 m intervals, and velocity vectors remained at 1 m.
8. Figure 6 a,b velocity values, represent the depth-averaged velocity, this is the River2D output. Yes, if these were bed or boundary values, they would be quite strange. The text and figure caption refer to depth-averaged velocities.
9. We are keeping this section (Ecosystem Services) as this establishes important context and contribution to field.
Round 2
Reviewer 1 Report
Accept as it is
Reviewer 4 Report
Needful is done in revised version, now I accepted as is
Reviewer 5 Report
Dear authors,
thanks for the corrections and the answers to my question.